# Heat-Treated Micronized Polyethylene Powder for Efficient Oil/Water Separating Filters

**DOI:** 10.3390/ma13143160

**Published:** 2020-07-15

**Authors:** Yasmin A. Mehanna, Colin R. Crick

**Affiliations:** 1Materials Innovation Factory, Department of Chemistry, University of Liverpool, Liverpool L69 7ZD, UK; Yasmin.Mehanna@liverpool.ac.uk; 2School of Engineering and Materials Science, Queen Mary University of London, Mile End Road, London E1 4NS, UK

**Keywords:** superhydrophobic, polyethylene, micronized powder, oil/water separation

## Abstract

The targeted separation of oil/water mixtures is a rapidly growing field of research, mainly due to contaminated water becoming an increasingly important environmental issue. Superhydrophobic materials are highly suited to this application; however, growing efforts are being devoted to developing applicable technologies within a range of research communities. The optimal technical solution is one that combines a high separation efficiency with a straightforward fabrication procedure at a low cost. In this report, micronized polyethylene powder has been utilized as a low-cost hydrophobic material to manufacture easy-to-fabricate filters. The effect of heating and solvent addition on the water repellence behaviour has been investigated, according to which the optimum fabrication conditions were determined. The filters show high water repellence (WCA = 154°) and efficient oil/water separation (~99%). The filters are designed to provide a readily achievable approach for the separation of oils (hydrophobic solvents) from water in a range of potential applications.

## 1. Introduction

Oil contaminated water is a significant environmental hazard that has a huge effect on the marine ecosystem and can cause an additional threat to many biological species [1,2]. The release of oil into the environment can result from waste produced by many industries (e.g., food, petrochemicals, and foundries), or leakage during oil drilling/transportation. As there is a growing reliance on these industries with increasing demand, in combination with the failure to significantly cut contamination in water waste, water purification is becoming an increasingly important field of research [1]. Numerous reports in the literature are aimed at fabricating a wide range of materials utilizing different designs and technologies to achieve this goal. Many designs have been reported to separate oil from water. These include oil skimmers [3], sponges [4,5,6], meshes [7,8], foams [9,10], and selective filters [11,12]. The common feature in all these designs is the ability for materials/devices to selectively interact with either oil or water. This leads to that component being absorbed and separated, while the other solvent is unselected [13]. This is generally achieved by controlling the relative hydrophobicity of the material, and therefore liquid selectivity can be achieved. 

High oil/water separation efficiencies have been widely achieved for various materials, with many exceeding a 97% efficiency value [14,15,16]. A common approach utilises metallic meshes, as they provide mechanically strong porous structures and can be treated to control their wetting properties [17]. This includes the controlled growth of ZnO on stainless steel meshes to create superhydrophobic and superoleophilic filters, where both high water contact angle (WCA, ≥152°) and high separation (above 99%) were achieved [18,19,20]. Textiles have been investigated due to their desirable low-cost, light-weight, and inherently rough microstructure, where the polymer coating of fibres is reported as a surface-modification approach [21,22]. Selective absorbent materials with a high proportion of oil intake have also been developed in many forms and utilizing different materials, including carbon nanotube aerogels [17], graphene oxide nanosheets [23], and polymer sponges [24]. In addition, films fabricated using heat-treating hydrophobic polymer powder have been reported to efficiently separate oil from water in harsh conditions [25]. For these examples, although an efficient separation could be obtained, other challenges hinder the potential applicability of these designs. Metallic meshes can involve using high-cost and heavy materials, along with the complexity of the controlled oxidation process for roughening the mesh surface [17]. With textiles, although they would seem a good alternative for meshes, they suffer from the low adhesion of physically-deposited coatings, which make using chemical functionalization routes a necessity, limiting the possible fabrication methods [17]. Oil absorbents have limited absorption capacity, which leads to a huge amount of required material when scaling-up the separation process [17]. When polymer powder is heated, the produced films are no longer permeable, and approaches to induce porosity are required to allow solvent passage [25]. Research is continuing to optimise these systems, considering their drawbacks and limitations to reach an optimal separation system. 

Superhydrophobic surfaces are commonly utilised for oil–water separation, as they generally have high water repellence and strong oil attraction. To construct a superhydrophobic surface, two main components should be present: (i) inherently hydrophobic surface chemistry, and (ii) a highly rough surface morphology [26]. This has been achieved via numerous methods reported in the literature, but generally involve either the roughening of an already hydrophobic material [27,28,29], or the hydrophobic treatment of a roughened material [30,31,32]. The use of nanoparticles as a scaffold for hydrophobic polymers has been widely reported [33,34]. Conversely, the controlled growth/deposition of a hydrophobic material to induce surface roughness has also been used [35,36]. Generally, superhydrophobic surfaces have features ranging from the nanoscale to a few microns [37]. Utilising this principle, a surface constructed from microparticles of a hydrophobic polymer would be sufficient to observe a superhydrophobic behaviour, as such particles being present on the surface would generate the required roughness. Micropowders of hydrophobic polymers have been reported for surface modification, which resulted in changing the wetting properties of the materials decorated with these particles [38,39].

As an example, polyethylene (PE) is a hydrophobic polymer, with a WCA of ~100° for a flat polymer layer deposited on a flat surface [40]. Owing to its compatibility with many systems, it has been reported numerously for the fabrication of superhydrophobic coatings [33,41,42]. The heat-treatment of PE powder to produce a hydrophobic film has been reported previously [25]. However, the high temperature applied resulted in a complete melting of the polymer, generating a flat, nonporous film that needs to be roughened and pricked to make it functional. Herein, we report a readily-permeable, rough-surface filter system for oil/water separation that uses PE micropowder. When heat-treated, it produces a solid permeable material that shows superhydrophobic properties and the ability to allow passage of oil, but impedes water. This was achieved by applying specific treatment conditions, which led to the heating being effective to bind the powder together without blocking possible pathways for liquid passage. The system consists of two layers fabricated from the same PE microparticles (Figure 1a–c). The main difference between them is the heat treatment that influences the binding mechanism and the resultant surface properties. For the first layer, an amount of PE micropowder is heat-treated at high temperature (140 °C), followed by the second layer which is spray-coated on top of the first layer, followed by another heat-treatment step (114 °C) (Figure 2). The prepared filter showed high hydrophobicity (up to a WCA of 154°) and high separation efficiency (up to 99%) was achieved for the separation of water/hydrophobic solvent mixtures. Compared to other designs, the reported materials are shown to provide an effective, versatile, and relatively straightforward means of achieving water purification. In addition, applying polymer in micronized-powder form satisfies the need for a low-cost material and enables the application of recycled polymer waste. Finally, they demonstrate the potential for immediate application for water treatment processes and oil-spill clean-ups.

## 2. Experimental

### 2.1. Materials

Finely-micronized PE with different mean particle sizes (three grades; 4.25–4.75, 5–7, and 7–9 µm, product codes are: MPP-620XXF, MPP-620VF and MPP-620F, respectively) were generously gifted by Kromachem Ltd. (Watford, UK), Chloroform, toluene (both analytical grade, ≥99.8%), dichloromethane (DCM, laboratory-grade, ≥99%), and hexane (HPLC grade, ≥95%) were purchased from Fisher Scientific. Loughborough, UK, Methylene blue and Nile Red (9-(Diethylamino)-5Hbenzo[a]phenoxazin-5-one) were purchased from Sigma Aldrich, Gillingham, UK, and Tokyo Chemical Industry UK Ltd., Oxford, UK, respectively. Filter papers (Ø—4.25 cm, grade 1, pore size 11 µm) were purchased from Whatman. 

### 2.2. Filter Preparation

The preparation method is illustrated in Figure 2. PE filters were fabricated using a dual-layer design prepared in two different steps. The first layer used 3.2 g of a PE powder, which was added into a Buchner funnel (Ø—5 cm) on top of a filter paper to prevent PE particles from falling through into the Buchner funnel frit. After roughly dispensing the powder in the paper, 10 mL of hexane was used to wet and more evenly distribute the PE across the surface using a Pasteur pipette. The funnel was then placed upright in a preheated oven at 140 °C for 10 min when the polymer was lightly compacted (using the base of a glass beaker and while the powder is still hot) this helped reduce cracking, and the formation of a uniform surface. To prevent the PE from sticking to the glass, the beaker was covered with a sheet of synthetic rubber. The funnel was left at the same temperature for another 5 min after compacting. 

For the second layer, 1 g of the same PE powder was dispersed in 40 mL of hexane. The entire solution was evenly spray-coated on top of the first layer, prepared previously. Spray-coating was carried out using a compression pump and airbrush gun (Voilamart) at a pressure of 2 bar. All spraying was carried out approximately 4 cm away from the surface. After spray coating was complete, another 10 mL of hexane was sprayed across the PE to ensure all powder had been placed uniformly and no powder was left at the spray nozzle. The funnel was returned to the oven at a lower temperature (114 °C) for 10 min. 

Comprehensive material analysis was hindered when the PE was formed inside the Buchner funnel. As a result, a sample analysis of the filter surface was mimicked on a glass microscope slide (Thermoscientific, 76 × 26 mm) by using the conditions outlined above for the secondary layer. Microscope slides were utilised for sample imaging, water contact angle (WCA), and tilt angle (TA) measurements, and compared to measurements taken using the filters fabricated within the Buchner funnels.

### 2.3. Separation Efficiency

Separation efficiency was examined using hydrophobic solvents (chloroform, dichloromethane (DCM), hexane, and toluene) and water. For each test, 1 mL of each of the hydrophobic solvent was mixed with 1 mL of water. All hydrophobic solvents were coloured with Nile-Red dye, and methylene blue was used to dye the water (a dye concentration of ≈7 mg/mL was used). This insured the visualisation of the separation process and an immediate indication of the purity of the solvent that passed through/collected on the filter. The separation efficiency was monitored by comparing the quantity of water applied to and then collected from the filter—this was determined using the difference in weight before and after separation. Between each test, the filter was heated for a few seconds using a heat gun and dried with a tissue to ensure none of the residual solvent/water was present. The reported efficiency percentage is the average of five readings taken for the same filter and solvent.

### 2.4. Characterization

A Kruss (DSA100E) Drop Shape Analyser was used to measure the tilt angles, for both the PE filters and the materials deposited onto glass microscope slides. These measurements were repeated at least ten times for each sample and the average was calculated. For WCA measurements conducted on the filters (in the Buchner funnel) and coated glass slides, water droplet volumes of 20 µL and 5 µL were used, respectively. For PE filters, WCAs were measured by taking photographs of water droplets placed on top of the PE filters while in the Buchner funnel. 

Water bouncing was done using the same glass funnels, and water droplets were dropped from a height of 20 mm (tip to the surface) using a micro-syringe fitted with a 27-gauge dispensing tip [43]. The water droplets from this tip were estimated at 8 μL in size and were left to detach under using their weight. Methylene blue was added to the water to aid visualization; this was not observed to change the behaviour of the water droplets on the surface. The bouncing was filmed at 500 frames per second using a SONY RX10-IV camera (SONY, Tokyo, Japan). Scanning electron microscopy (SEM) images were performed using a Hitachi S4800 microscope (Hitachi, Tokyo, Japan) operating at an acceleration voltage of 3–5 kV for polymer samples deposited on glass slides. Samples were coated with a thin layer of chromium metal to ensure the conductivity and high-quality SEM images.

## 3. Results and Discussion

The PE micropowders were chosen as they are inherently hydrophobic and rough. In addition, PE is a thermoplastic. All of the PE powders were exceptionally hydrophobic as received (≈8 bounces using water bouncing test, corresponding to a WCA ≈ 165°) [43], and this was due to the combination of their hydrophobic chemistry and roughness on the correct length scales. However, the particles could not be used as received to form a consistent superhydrophobic surface, nor as a water-oil separation membrane, as the microparticles were mobile when impacted by liquids. To prevent this, heat treatment of the PE powders would be used to force the particles to agglomerate, forming a continuous membrane for separation that would be fixed in place. The thermoplastic nature of the PE would be utilised to induce partial melting and bind the particles together at their edges, thus forming a permeable membrane that would repel water and allow the passage of hydrophobic solvents. 

Although the loose PE powder is characterized by a high hydrophobicity, this hydrophobicity could decrease dramatically with excessive heating (as low as 109° when heated at 160 °C). Therefore, the choice of heat treatment conditions (i.e., the type/quantity of solvent used for particle dispersion, the temperature used, and exposure time) should be carefully considered as this has a large impact on the manufactured filter properties. This was thoroughly investigated in order to establish the optimal process for generating elevated hydrophobicity, and thus high oil-water separation efficiencies. 

Heat-treatment of the PE filters without dispersal of the powder (using a solvent) proved incapable of generating a solid and uniform material. This resulted in a low powder cohesion, and in most cases, there was no (or little) physical change in the material form, regardless of how long the heat-treatment lasted. This was not observed when the powder was dispersed with the solvent prior to heat treatment, as after 15 min (at 140 °C) the powder started to adhere together to form particle agglomerates. This suggested that using a solvent to disperse the PE enhanced the agglomeration process by evenly distributing the microparticles, which led to more consistent heating throughout the material and a regular (partial) melting behaviour resulting in a uniform filter.

Two types of solvent were examined for pre-treating the microparticles (hexane and toluene). These were chosen for their ability to disperse PE particles without dissolving them. As the wetting results in a greater level of particle adhesion, the time of solvent exposure was probed. Giving the high difference in the boiling point between the two solvents (hexane: 68 °C, toluene: 110 °C), this provided a difference in evaporation time, which in turn affected the resultant filter. Hexane was found to be most suitable, as its boiling point was low enough to ensure rapid evaporation, compared to toluene, which required a longer time for complete evaporation. The filters formed using toluene resulted in a less hydrophobic base layer, and this was assumed to be caused by a flatter morphology. When hexane was used, a greater level of surface roughness was preserved, leading to a degree of superhydrophobicity, which was indicated by four water bounces (corresponding to a WCA around 155 °C) [43]. The difference in bouncing behaviour is indicated in Figure 3. 

The quantity of solvent used was also found to affect the surface roughness. Using a large quantity of hexane (≥4 mL/g) would lead to the same effect as using toluene, as the powder would remain wetted for a long time and evaporation would be delayed. With too little solvent quantities (≤2 mL/g) this was found to limit the extent that the heat-treatment would generate a solid/uniform structure. To minimize the loss in hydrophobicity while maintaining a well-structured filter, the quantity of solvent that would be just enough to wet the whole powder was used. In addition, the solvent was poured slowly using a Pasteur pipette to ensure even distribution and to avoid putting too much solvent. Typically, for a 3.5 g of powder, 10 mL of hexane was enough to reach a good wetting. The wetting with hexane was not observed to have an impact on the overall PE particle size (see Appendix A).

The ideal temperature of heat treatment was explored, using the reported PE bulk melting point (114 °C) as a foundation. Temperatures lower than this would not be sufficient for inter-particle binding, while when heated at 160 °C, a large portion of the powder would melt completely after 10 min. This high-temperature treatment led to a non-porous, marginally hydrophobic material when cooled to room temperature (Figure 4j–l). Holding the polymer at its melting point was found to be not enough to form a well-structured filter (i.e., poor structural integrity), despite the good superhydrophobicity observed (WCA above 150 °C). Therefore, a high temperature, typically 140 °C, was utilised to form a rigid material, which is still permeable so the hydrophobic solvent can pass through. The highest measured WCA at 140 °C (achieved with the S-µ particles) was found to be 142°.

SEM was used to examine how temperature influenced the powder microstructure and inter-particle binding (Figure 4). For loose powder (Figure 4a–c), as expected, particle binding was not observed. Heat-treating at the powder melting temperature (114 °C, Figure 4d,f) shows a moderate degree of cohesion and inter-connectivity. Retention in the feature roughness can be observed, where microscale features remain present. Moving to a higher temperature (140 °C, Figure 4g–i) a large degree of particle agglomeration can be seen. In addition, small (~200 nm) wrinkles can be observed on the surface of the polymer particles, indicating a high degree of surface melting and resultant resolidification [26]. This higher level of surface melting is observed on the macroscale with a greater level of cohesion in the filter material. Operating at a higher temperature (160 °C, Figure 4j–l) results in a complete melting of the polymer. This led to the loss of the initial morphology from the microparticles, resulting in a structure closer to a flat PE film. Atomic force microscopy (AFM) was considered for surface morphological analysis, however, the large features would exceed that of the maximum possible z-travel during an AFM scan.

The initial layer of heat-treated particles provided a permeable material with relatively high WCAs (142°, heated at 140 °C); however, these were not yet superhydrophobic. To increase the WCAs for the filters, a lower temperature heat treatment was required. It was found that a second layer of powder suspension, sprayed onto this initial layer and heated at the PE melting temperature (114 °C), combined both integrity with superhydrophobicity (WCA > 150°). The base layer provides rigidity, while the second layer gives high water repellence (Figure 1). The higher WCAs were attributed to a higher surface roughness provided by the lower temperature treatment. 

Hydrophobicity was accessed for the prepared filters. As highlighted previously, the wetting behaviour of the PE filter was highly dependent on the heat treatment process. This is illustrated by high-temperature treatment (160 °C for 10 min) which results in a loss of the superhydrophobic nature of the loose powder, where WCAs decreased to 113° (Figure 5e). WCA measurements were not straightforward when the filter was mounted in the Buchner funnel. This was due to optical distortions caused by the thick glass of the Buchner funnel, in addition to the high surface roughness (macro/microscale) that made full visualisation of water droplets challenging (Figure 5f). This difficulty stemmed from small polymer features obscuring the three-phase point in the water droplet images. Therefore, wetting analysis of the filters was obtained through three key alternative measurements; (i) WCAs were taken for spray-coated PE layers on glass microscope slides (Figure 5a–e), (ii) water droplet tilt angles on the fabricated filters and coated glass slides, and (iii) water bouncing measurements (Figure 5g). The major contributor of the high WCAs is the top PE layer; in replicating these coatings on glass slides, an unhindered approximation of the WCAs of would be possible. For the largest particle size tested (L-µ powder), the particle dispersion was too large to be sprayed, and most of the polymer could not pass through the spray nozzle. As a result, a superhydrophobic surface was not achieved for this powder size. For the S-µ and M-µ powder, WCAs of 154° and 153° were measured respectively. In addition, tilt angles of 11° and 13° were achieved for the S-µ and M-µ filters respectively, which were higher than the same measurements made on PE coatings on glass slides (8°/6° respectively). A full list of the filter tilt angels is included in the Appendix A. The difference in the constructed filters and glass slide filters is hypothesised through the additional macro-roughness achieved in the filter samples that enabled greater pinning of water droplets. Overall, the S/M-µ filters were judged to have similar wetting behaviours.

Water bouncing was the final wetting characterisation measurement. The uneven surfaces made the measuring of WCAs and tilt angles challenging, and water bouncing provided a method to overcome this. The water bouncing measurements can indicate static wetting properties (akin to the WCAs), but also demonstrate the dynamic wetting/de-wetting behaviour relevant to oil-water separation applications. After few bounces (≈5), the droplet travels across the surface of the PE filter, and tends to become adhered to the un-coated glass part of the funnel (Figure 5g). The measurements show clearly the water-repelling nature of the filter material, with water bouncing proving an estimated WCAs of the filter membranes around 160° [43]. A video for water bouncing can be found in the Appendix A.

Separation efficiency experiments were carried out by depositing water/solvent mixtures onto the filter material (Figure 6a). Typically, 1.00 g (±0.01) of water was weighed and transferred by a syringe, then around 1 mL of hydrophobic solvent (chloroform, DCM, hexane, or toluene) was taken up into the same syringe. The entire syringe contents were emptied onto the filter all at once, with an approximate distance of 3 cm. After a few seconds, the water on the filter surface was collected via syringe and weighed. The separation efficiency percentage was calculated by dividing the mass of the water collected by the mass of the water poured onto the filter. A recording for the separation of chloroform/water mixture using an S-µ PE filter could be found in the Appendix A, and an image of a chloroform/water mixture separated using a S-µ PE filter is shown (Figure 6c,d). This larger-scale experiment enabled the estimate of flux (~23 mL s^−1^ m^−2^); however, this could be improved using an applied pressure, or variation in the filter design (e.g., thickness, particle diameter, etc.…).

The separation efficiency for all solvents and PE filters is shown in Figure 6b. Generally, the separation was shown to be very efficient, with an average of 98.9% for all experimental runs. A maximum of 99.9% was recorded for the separation of water/chloroform mixture using the S-µ particle size, and a minimum of 98.1% was recorded for separating the same mixture using the M-µ particle size. As noted earlier, the spraying of L-µ particles was non-optimal, and hence the filters made using that size showed reduced WCAs. However, they tended to still perform well. This suggested that for these materials, superhydrophobicity was not essential to achieving high separation efficiencies. For the smallest particle size used (S-µ), some readings were above 100%. This was expected to be due to an experimental error in the measurement of the mass of recovered water. Extra mass generated from recovered oil was deemed to be less likely, as red colouration from the dyed oil was not observed. Overall, the filters are found to be well suited to oil-water separation. A recorded video for separation of water/hexane mixture using an S-µ filter could be found in the SI. Although the L-µ filters highlighted that superhydrophobicity was not necessary for high separation efficiency, this is expected to be beneficial for a potential application, particularly if recovery of the water is important, as superhydrophobicity would aid the removal of water from the filter surface. Each filter was used to separate four different solvent/water mixtures, with a minimum repeat of 5 times per mixture type. Therefore, each filter was able to carry out no less than 20 separations while maintaining a consistently high separation efficiency (error bars shown in Figure 6b). Physical resilience of the filters (e.g., to abrasion) was not comprehensively analysed, but they were able to show moderate resistance to manual handling, although high physical strength was not a major focus during their development. Given this, it is still expected that the PE filters are generally applicable, and show high potential for future development. An example of potential utilisation is the separation of emulsified systems, particularly for those with water droplets that remain larger than the pores of the permeable membrane (~1 µm), and thus are able to be separated from an oil phase.

## 4. Conclusions

The reported method utilises micronized PE in the formulation of the dual-layer filter, generated using a straightforward heat-treatment process to form a ridged filter with a rough surface. The fabrication conditions (including temperature and dispersion solvent type/quantity) were thoroughly investigated, along with different sizes of PE microparticles. The two layers were used to combine both structure integration and high surface roughness. The filters show high hydrophobicity and efficient separation of different oil/water mixtures, which were as high as 99.9% for chloroform/water mixtures separated using a PE microparticle average size of 4.5 µm. This material design demonstrated in this research is aimed to offer an approachable and cost-effective method to achieve the efficient separation of oil/water mixtures. The filter fabrication approach demonstrates a high potential for versatility, and resultant utilisation in a real-world application, in both academic research and industry.

## Figures and Tables

**Figure 1 materials-13-03160-f001:**
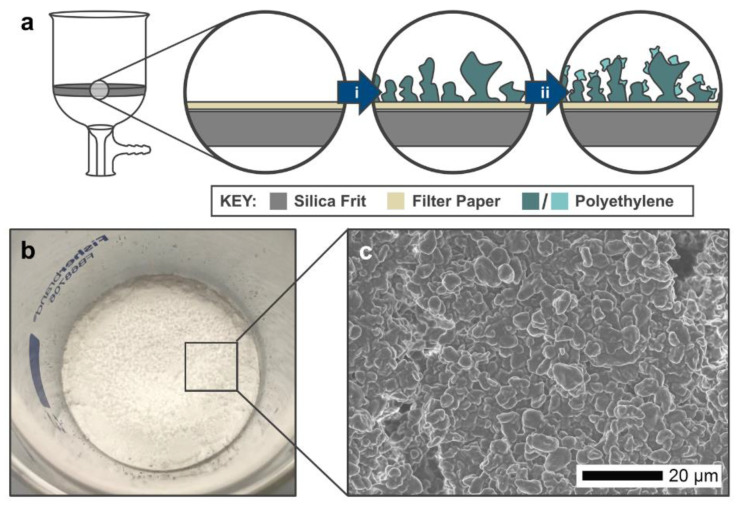
(**a**) A schematic showing the composition of the PE micropowder filters. The filter is constructed using a Buchner funnel/silica frit with a filter paper placed on top of the frit. The first layer (**i**) uses a heat treatment temperature of 140 °C, which forms a uniform base for the second layer (**ii**). The second layer uses a lower heat treatment (114 °C) which preserves the high surface roughness. (**b**) A digital photo of the filter made using S-µ sized PE powder. (**c**) An SEM image of the S-µ powder heated at 114 °C.

**Figure 2 materials-13-03160-f002:**
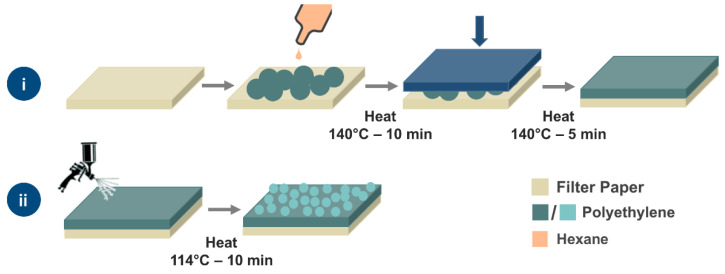
A schematic showing the fabrication method of the PE micropowder filters. For the primary layer (**i**), the PE micropowders are placed on a filter paper, dispersed by hexane and heat-treated (140 °C), followed by compacting and further heating. The second layer (**ii**) of PE is deposited by spray coating a suspension of particles in hexane and is annealed at 114 °C.

**Figure 3 materials-13-03160-f003:**
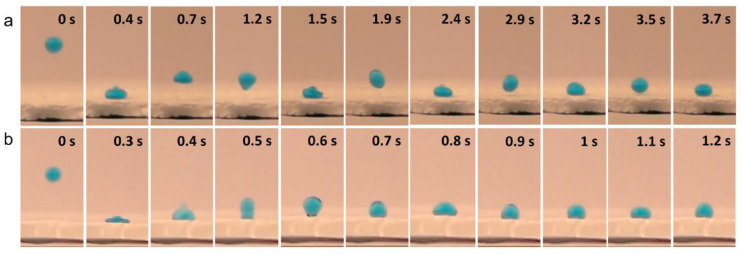
Photographs of an 8 μL water droplet (coloured with methylene blue) which were dropped from a height of 20 mm (tip to surface) and left to bounce on PE surfaces cured at 110 °C, using (**a**) hexane and (**b**) toluene as a dispersing solvent.

**Figure 4 materials-13-03160-f004:**
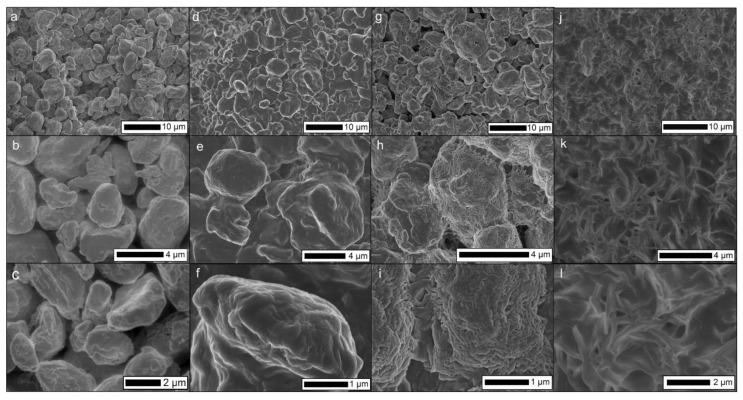
SEM images of S-µ PE powder under different heat treatments, where (**a**–**c**) shows a loose powder without heat treatment, and with heat treatment at 114 °C (**d**–**f**), 140 °C (**g**–**i**), and 160 °C (**j**–**l**).

**Figure 5 materials-13-03160-f005:**
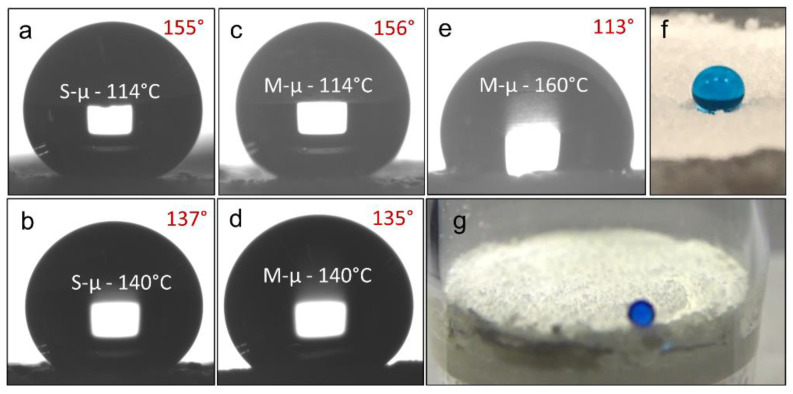
(**a**–**e**) Images showing WCAs for the PE secondary layer sprayed on a glass slide. The powder sizes and heat-treatment temperatures are: S-µ powder heated at (**a**) 114 °C, (**b**) 140 °C, and M-µ powder heated at (**c**) 114 °C, (**d**) 140 °C and (**e**) 160 °C. (**f**) A 20-µL water droplet on an S-filter mounted in a glass funnel where the PE dual-layer filter is placed. Besides light distortion through glass, some polymer features are covering the liquid/air interface at the filter surface, making accurate measurements of CAs challenging. (**g**) An image from bouncing experiment on a PE filter.

**Figure 6 materials-13-03160-f006:**
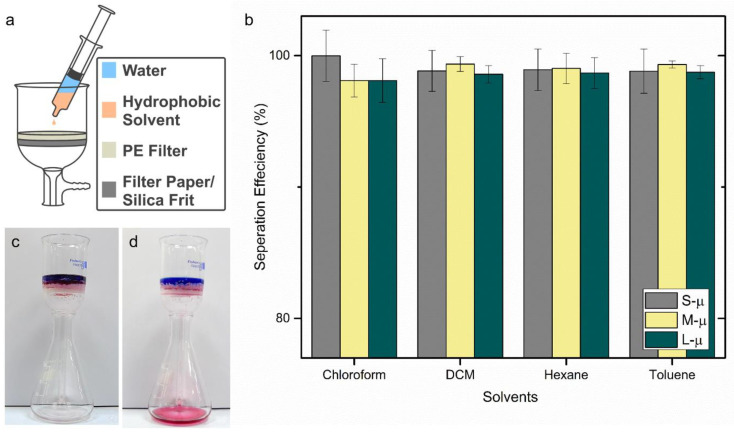
(**a**) The separation efficiency measurement setup. (**b**) The separation efficiency results for different water/oil mixtures with filters made using different PE powder sizes. (**c**,**d**) The chloroform/water mixture before (**c**) and after (**d**) separation using an S-µ PE filter.

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
