# Peer review of "Heat-Treated Micronized Polyethylene Powder for Efficient Oil/Water Separating Filters"

_materials, 2020, doi:10.3390/ma13143160_

Round 1
Reviewer 1 Report
The authors tried to provide an easy and low cost method to elaborate a membrane for oil/water separation. The use of PE follow some already published works. The spray drying method make the process easier. They reported the effect of the temperature on the contact angle and reached a superhydrophobic surface with optimal formulation. The article could be published after clarifying some points. Somme recommendations should be adressed to improve the manuscript: The authors should provide the size of the PE particles obtained after the spray driyng and provide, if relevent, the effect of the solvant in the size. Roughness measurement should be provided for each sample and at variable temperature To support the conclusion, the authors are encouraged to provide the sliding contact angle. What is the stability of this membrane, when used several time? Figure 5: the separation efficiency. What is the time needed to separate each oil/xater mixture? A figure with variable time scale for the separation is highly recommanded.
Reviewer 2 Report
This work reported the design of superhydrophobic materials for Oil/Water separating filters based on the Polyethylene powder heat-treatment.
There are some points should be treated as follows:
1- The abstract part should cover the object, main results obtained, approach, and the work novelty. Therefore, it is highly recommended to revise the provided abstract with focusing on the aim, merits of approach and materials, as well as novelty needs more clarifications.
2- the author should provide three keywords.
3- the introduction cover the work but there is a limitation and lack of background related to the materials used, developing on the problem and challenges, and the advantage of the current approach.
In addition the author should provide clear claim that cover the aim, materials, and novelty at the end on introduction.
Figure 1 needs more information about the materials, design, and its morphological structure, as with as the amount if possible.
4- regarding to the experimental section: this part needs to revise carefully. Where, it is recommended to describe all steps as schematic design merged within figure 1. This figure should cover all steps.
5- the author provided SEM analysis, but it is highly recommended to provide more SEM figures with high magnification and scale bar should be clear.
The author has mentioned porous morphology many times, what about the surface analysis such N2 Adsorption isotherm, as well as other analysis such as AFM, FTIR,...? before and after using?
What about using real samples?
Reviewer 3 Report
This manuscript describes fabrication, characterization and application of non-wetting filters made by depositing fine micronized PE powder on filter paper fixed on a support. The authors then demonstrate that the after some thermal treatment and pressing part of the powder into the filter paper texture, they could separate solvents or oil like liquids from water by gravity decantation. The manuscript has certain points of interest for the readers of the journal materials, however in the interest of improving the manuscript quality further, the authors are invited to address some questions, comments and concerns listed below:
- Please comment or if possible demonstrate how robust the filers are like whether they would resist say mild abrasion or bending folding attempt. This is important for practicality. A demonstration of the degree of certain durability level is needed.
- What about use cycles? How many times can these filters be used? One time only? Please execute some experiments/tests to this effect (separation cycling).
- A few earlier works demonstrated fabrication of non-wetting surfaces by depositing or attaching sub-microscopic hydrophobic polymer particles on certain surfaces (without using any solvents or wet chemistry). For the sake of completeness and due credit, in the introduction part, these works should be discussed (even though they were not made for oil separation): Colloid and Polymer Science 291.2 (2013): 367-373 and Materials Today Communications 3 (2015): 57-68.
- Please provide cross section SEM images as well for selected filters.
- Please also comment or even demonstrate if oil-water mixtures in the forms of emulsions may be separated with this material system. Many real oil polluted water problems facing industry go toward emulsions rather than well-separated liquids.
Round 2
Reviewer 2 Report
The paper presents in its revised form the efforts made by the author, therefore I recommended to be accepted after a few minor adjustments needed as follows:
In Figure 2: It is recommended that more information and details be added to the caption to describe the figure well and assist readers
The author asked about the meaning of "real samples", real samples means natural contaminated samples.
In addition, after making changes there is no need to return the manuscript for review, it can be accepted directly without further review.
Author Response
The caption of Figure 2 has been updated.
The use of contaminated samples has not been attempted, if deemed necessary, the authors would need more specific information to complete further testing. However, additional time would be needed, due to current limited access to laboratory facilities.
Reviewer 3 Report
The revisions and clarifications appear to be adequate. The authors stated that due to lock down regulations, full time operations have been limited in their labs. Nonetheless, technically speaking, the work is sound and the revised version may now be published in the journal Materials.
Author Response
We thank the referee for their time in reviewing the manuscript and understanding during these challenging times.